# PASTA: PRETRAINED ACTION-STATE TRANSFORMER AGENTS

## ABSTRACT

Self-supervised learning has brought about a revolutionary paradigm shift in various computing domains, including NLP, vision, and biology. Recent approaches involve pre-training transformer models on vast amounts of unlabeled data, serving as a starting point for efficiently solving downstream tasks. In the realm of reinforcement learning, researchers have recently adapted these approaches by developing models pre-trained on expert trajectories, enabling them to address a wide range of tasks, from robotics to recommendation systems. However, existing methods mostly rely on intricate pre-training objectives tailored to specific downstream applications. This paper presents a comprehensive investigation of models we refer to as pre-trained action-state transformer agents (PASTA). Our study uses a unified methodology and covers an extensive set of general downstream tasks including behavioral cloning, offline RL, sensor failure robustness, and dynamics change adaptation. Our goal is to systematically compare various design choices and provide valuable insights to practitioners for building robust models. Key highlights of our study include tokenization at the action and state component level, using fundamental pre-training objectives like next token prediction, training models across diverse domains simultaneously, and using parameter efficient fine-tuning (PEFT). The developed models in our study contain fewer than 10 million parameters and the application of PEFT enables fine-tuning of fewer than 10,000 parameters during downstream adaptation, allowing a broad community to use these models and reproduce our experiments. We hope that this study will encourage further research into the use of transformers with first-principles design choices to represent RL trajectories and contribute to robust policy learning.

## 1 INTRODUCTION

Reinforcement Learning (RL) has emerged as a robust framework for training highly efficient agents to interact with complex environments and learn optimal decision-making policies. RL algorithms seek effective strategies by maximizing the cumulative rewards obtained from interactions with the environment, leading to remarkable achievements in diverse applications, ranging from game-playing to robotics (Silver et al., 2014; Schulman et al., 2016; Lillicrap et al., 2016; Mnih et al., 2016). These algorithms often comprise multiple components that are essential for training and adapting neural policies. For example, model-based RL involves learning a model of the world (Racanière et al., 2017; Hafner et al., 2019; Janner et al., 2019; Schrittwieser et al., 2020) while most model-free policy gradient methods train a value or Q-network to control the variance of the gradient update (Mnih et al., 2013; Schulman et al., 2017; Haarnoja et al., 2018; Hessel et al., 2018). Training these multiple networks is challenging due to their nested nature (Boyan & Moore, 1994; Anschel et al., 2017) and the need to extract meaningful state-action space features along with relevant credit assignment in complex decision-making problems. Consequently, these factors contribute to fragile learning procedures, high sensitivity to hyperparameters, and limitations on the network's parameter capacity (Islam et al., 2017; Henderson et al., 2018; Engstrom et al., 2020).

To address these challenges, various auxiliary tasks have been proposed, including pre-training different networks to solve various tasks, such as forward or backward dynamics learning (Ha & Schmidhuber, 2018; Schwarzer et al., 2021) as well as using online contrastive learning to disentangle feature extraction from task-solving (Laskin et al., 2020; Nachum & Yang, 2021; Eysenbach et al., 2022). Alternatively, pre-training agents from a static dataset via offline RL without requiring

interaction with the environment also enables robust policies to be deployed for real applications. Most of these approaches rely either on conservative policy optimization (Fujimoto & Gu, 2021; Kumar et al., 2020) or supervised training on state-action-rewards trajectory inputs where the transformer architecture has proven to be particularly powerful (Chen et al., 2021; Janner et al., 2021).

Recently, self-supervised learning has emerged as a powerful paradigm for pre-training neural networks in various domains including NLP (Chowdhery et al., 2022; Brown et al., 2020; Touvron et al., 2023), computer vision (Dosovitskiy et al., 2020; Bao et al., 2021; He et al., 2022) or biology (Lin et al., 2023; Dalla-Torre et al., 2023), especially when combined with the transformer architecture. Inspired by impressive NLP results with the transformer architecture applied to sequential discrete data, most self-supervised techniques use tokenization, representing input data as a sequence of discrete elements called tokens. Once the data is transformed, first-principles objectives such as mask modeling (Devlin et al., 2018) or next token prediction (Brown et al., 2020) can be used for self-supervised training of the model. In RL, recent works have explored the use of self-supervised learning to pre-train transformer networks with expert data. While these investigations have yielded exciting outcomes, such as zero-shot capabilities and transfer learning between environments, methods such as MTM (Wu et al., 2023) and SMART (Sun et al., 2023) often rely on highly specific masking techniques and masking schedules (Liu et al., 2022a), and explore transfer learning across a limited number of tasks. Hence, further exploration of this class of methods is warranted. In this paper, we provide a general study of the different self-supervised objectives and of the different tokenization techniques. In addition, we outline a standardized set of downstream tasks for evaluating the transfer learning performance of pre-trained models, ranging from behavioral cloning to offline RL, robustness to sensor failure, and adaptation to changing dynamics.

**Our contributions.** With this objective in mind, we introduce the PASTA study, which stands for Pretrained Action-State Transformer Agents. This study provides comprehensive comparisons involving four pre-training objectives and two types of tokenization techniques, with multiple pre-training datasets and a collection of 23 downstream tasks, categorized into three groups and across three continuous control environments. The PASTA downstream tasks encompass imitation learning and standard RL to demonstrate the versatility of the pre-trained models. Moreover, we explore scenarios involving physical regime changes and observations alteration to assess the zero-shot performance and stability of the pre-trained models. We summarize the key findings of our study below:

1. **Tokenize trajectories at the component level.** Tokenization at the component level significantly outperforms tokenization at the modality level. In other words, it is more effective to tokenize trajectories based on the individual components of the state and action vectors, rather than directly tokenizing states and actions as is commonly done in existing works.

2. **Prefer first-principles objectives over convoluted ones.** First principles training objectives, such as random masking or next word prediction with standard hyperparameters match or can even outperform more intricate and task-specific objectives carefully designed for RL, such as those considered in MTM or SMART.

3. **Pre-train the same model on datasets from multiple domains.** Simultaneously pre-training the model on datasets from the three environments leads to enhanced performance across all three environments compared to training separate models for each environment.

4. **Generalize with a small parameter count.** All of the examined models have fewer than 10 million parameters. Hence, while these approaches are both affordable and practical even on limited hardware resources, the above findings are corroborated by experimentation with three transfer learning scenarios: a) probing (the pre-trained models generate embeddings and only the policy heads are trained to address downstream tasks), b) parameter-efficient fine-tuning (PEFT) (introducing a limited number of weights to the pre-trained model and fine-tuning them for solving downstream tasks), and c) zero-shot transfer.

## 2 RELATED WORK

**Self-supervised Learning for RL.** Self-supervised learning, which trains models using unlabeled data, has achieved notable success in various control domains (Liu & Abbeel, 2021; Yuan et al., 2022; Laskin et al., 2022). One effective approach is contrastive self-prediction (Chopra et al., 2005; Le-Khac et al., 2020; Yang & Nachum, 2021; Banino et al., 2021) which have proven effective

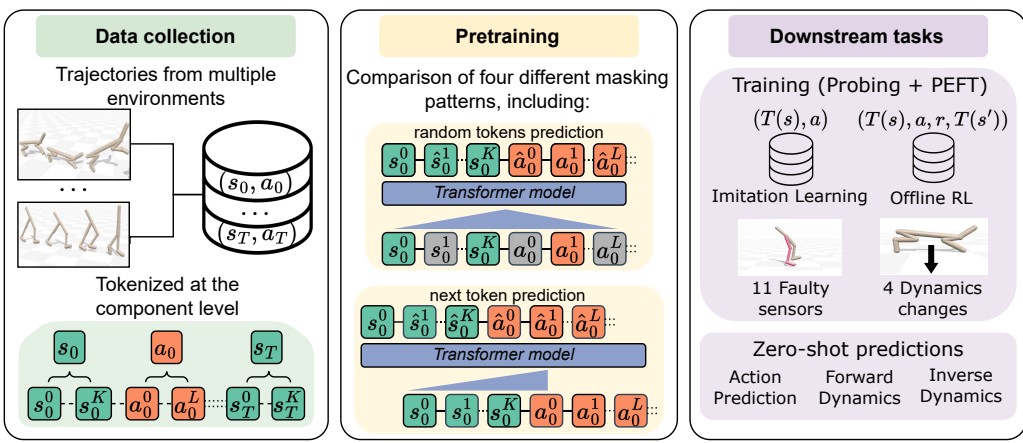

Figure 1: Illustration of the PASTA study. **Left:** State-action trajectories are collected from multiple environments and are tokenized at the component level. **Middle:** A transformer model is pre-trained by processing fixed-size chunks of these sequences. It learns latent representations $T(s)$ of the environments' states. In this study, we compare different tokenization schemes, masking patterns, and pre-training objectives, *e.g.*, random tokens prediction (BERT) or next token prediction (GPT). **Right:** The representations of the pre-trained transformer models are evaluated on multiple downstream tasks in which the learned representation $T(s)$ serves as a surrogate state for the policy.

in efficient data augmentation strategies, enabling downstream task solving through fine-tuning, particularly in RL tasks (Laskin et al., 2020; Nachum & Yang, 2021). Our study aligns with this trend, focusing on domain-agnostic self-supervised mechanisms that leverage masked predictions to pre-train general-purpose RL networks.

**Offline RL and Imitation Learning.** Offline learning for control involves leveraging historical data from a fixed behavior policy $\pi_b$ to learn a reward-maximizing policy in an unknown environment. Offline RL methods are typically designed to restrict the learned policy from producing out-of-distribution actions or constrain the learning process within the support of the dataset. Most of these methods usually leverage importance sampling (Sutton et al., 2016; Nair et al., 2020; Liu et al., 2022c) or incorporate explicit policy constraints (Kumar et al., 2019; Fujimoto & Gu, 2021; Fakoor et al., 2021; Dong et al., 2023). In contrast, Imitation learning (IL) focuses on learning policies by imitating expert demonstrations. Behavior cloning (BC) involves training a policy to mimic expert actions directly while Inverse RL (Ng et al., 2000) aims to infer the underlying reward function to train policies that generalize well to new situations. In contrast, the models investigated in PASTA focus on learning general reward-free representations that can accelerate and facilitate the training of any off-the-shelf offline RL or imitation learning algorithm.

**Masked Predictions and Transformers in RL.** Recently, self-supervised learning techniques based on next token prediction (Brown et al., 2020) and random masked predictions (Devlin et al., 2018) have gained popularity. These methods involve predicting missing content by masking portions of the input sequence. These first-principles pre-training methods have achieved remarkable success in various domains, including NLP (Radford et al., 2018; 2019), computer vision (Dosovitskiy et al., 2020; Bao et al., 2021; Van Den Oord et al., 2017), and robotics (Driess et al., 2023). We explore the effectiveness of different variants of these approaches, with various masking patterns and pre-training objectives, in modeling RL trajectories and learning representations of state-action vector components. Transformer networks have been particularly valuable for these purposes. The decision transformer (Chen et al., 2021) and trajectory transformer (Janner et al., 2021) have emerged as offline RL approaches using a causal transformer architecture to fit a reward-conditioned policy, paving the way for subsequent work (Zheng et al., 2022; Yamagata et al., 2022; Liu et al., 2022a; Lee et al., 2023). Notably, GATO (Reed et al., 2022) is a multi-modal behavioral cloning method that directly learns policies, while PASTA focuses on pre-training self-supervised representations. Additionally, MTM (Wu et al., 2023) and SMART (Sun et al., 2023) propose original masking objectives for pre-training transformers in RL. MTM randomly masks tokens while ensuring some

tokens are predicted without future context. It uses modality-level masking and is limited to single-domain pre-training. Conversely, SMART uses a three-fold objective for pre-training a decision transformer with forward dynamics prediction, inverse dynamics prediction, and "random masked hindsight control" with a curriculum masking schedule. It focuses on processing real-valued visual observation sequences and investigates generalization across different domains. In PASTA, we compare several first-principles pre-training objectives without a masking schedule to these state-of-the-art approaches across multiple environments and diverse downstream tasks.

## 3 THE PASTA STUDY

### 3.1 PRELIMINARIES

**Self-supervised Learning framework.** In this paper, we study self-supervised learning (Balestriero et al., 2023) techniques to pre-train models on a large corpus of static (offline) datasets from interactions with simulated environments, as done in Shah & Kumar (2021); Schwarzer et al. (2023). By solving pre-training objectives, such as predicting future states or filling in missing information, the models learn to extract meaningful features that capture the underlying structure of the data. We focus our study on the use of the transformer architecture due to its ability to model long-range dependencies and capture complex patterns in sequential data. In addition, the attention mechanism is designed to consider the temporal and intra-modality (position in the state or action vectors) dependencies. After pre-training the models, we evaluate their capabilities to solve downstream tasks. This analysis is done through the lenses of three mechanisms: (i) probing, (ii) parameter-efficient fine-tuning (PEFT) (Liu et al., 2022b; 2021; Hu et al., 2021), and (iii) zero-shot transfer. The goal of the study is to investigate which pre-training process makes the model learn the most generalizable representations to provide a strong foundation for adaptation and learning in specified environments. An illustration of the approach adopted in PASTA is given in Figure 1.

**Reinforcement Learning framework.** In this paper, we place ourselves in the Markov Decision Processes (Puterman, 1994) framework. A Markov Decision Process (MDP) is a tuple $M = \{\mathcal{S}, \mathcal{A}, \mathcal{P}, R, \gamma\}$, where $\mathcal{S}$ is the state space, $\mathcal{A}$ is the action space, $\mathcal{P}$ is the transition kernel, $\mathcal{R}$ is the bounded reward function and $\gamma \in [0, 1)$ is the discount factor. Let $\pi$ denote a stochastic policy mapping states to distributions over actions. We place ourselves in the infinite-horizon setting, *i.e.*, we seek a policy that optimizes $J(\pi) = \mathbb{E}_\pi[\sum_{t=0}^\infty \gamma^t r(s_t, a_t)]$. The value of a state is the quantity $V^\pi(s) = \mathbb{E}_\pi[\sum_{t=0}^\infty \gamma^t r(s_t, a_t) | s_0 = s]$ and the value of a state-action pair $Q^\pi(s, a)$ of performing action $a$ in state $s$ and then following policy $\pi$ is defined as: $Q^\pi(s, a) = \mathbb{E}_\pi[\sum_{t=0}^\infty \gamma^t r(s_t, a_t) | s_0 = s, a_0 = a]$.

### 3.2 TOKENIZATION

**Building Motivation.** Tokenization is a fundamental technique in self-supervised learning as it is an effective way to apply first-principles objectives for neural network pre-training. By representing states and actions as vector components and learning from diverse environments within the same physics engine, tokenization at this level enables representation models to reason effectively about interactions among different parts of a robot. This approach is expected to enhance performance across various morphologies and downstream tasks. For example, in a bipedal locomotion task, breaking down leg movements into thigh and shin components can facilitate the learning of more stable and efficient walking or running. We explore whether this granularity of tokenization offers a promising approach to address continuous control problems solely from a sequential perspective.

**Component-level Tokenization.** A key focus of the PASTA study is the representation of trajectories at the level of vector components for states and actions. Instead of considering each trajectory as a sequence of state, action (and often return) tuples, as done in most previous work, including SMART (Sun et al., 2023) and MTM (Wu et al., 2023), we break the sequences down into individual state and action *components*. Additionally, we exclude the return to develop a general method applicable to reward-free settings and learn representations that are not tied to task-specific rewards (Stooke et al., 2021; Yarats et al., 2021). This level of tokenization allows capturing dynamics and dependencies at different space scales, as well as the interplay between the agent's morphological actions and resulting states across different robotic structures. As we observe in Section 4, this results in more detailed representations that improve the performance of downstream tasks.

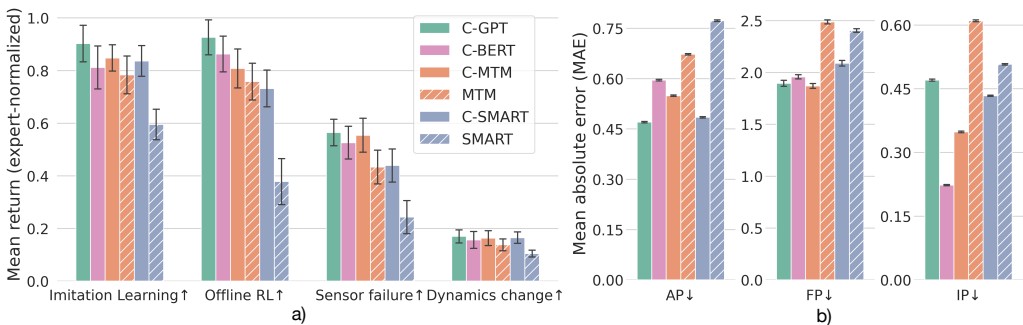

Figure 2: Performance aggregation of the PASTA pre-trained models (C-*) and modality-level models (MTM and SMART) with different masking and training objectives, evaluated on **a)** the representation learning tasks with fine-tuning and **b)** the zero-shot transfer tasks: Action Prediction (AP), Forward Prediction (FP), and Inverse Prediction (IP). Results are aggregated over all environments. We developed our own implementation of MTM and SMART using the same masking patterns and training objectives. ↑ (resp. ↓) indicates that higher (resp. lower) is better.

### 3.3 PRE-TRAINING

**Trajectory modeling.** The PASTA study includes different types of self-supervised learning strategies, each using different combinations of random token masking and/or next token prediction. Next token prediction uses autoregressive masking, while random masked prediction aims to learn from a sequence of trajectory tokens denoted as $\tau = (s_0^0, ..., s_0^K, a_0^0, ..., a_0^L, ..., s_T^0, ..., s_T^K)$. The model's task is to reconstruct this sequence when presented with a masked version $\hat{\tau} = T_\theta(\texttt{Masked}(\tau))$, where K is the observation space size, L is the action space size and T is an arbitrary trajectory size. Here, $T_\theta$ refers to a bi-directional transformer, and $\texttt{Masked}(\tau)$ represents a modified view of $\tau$ where certain elements in the sequence are masked. For instance, a masked view could be $(s_0^0, ..., s_0^K, a_0^0, ..., a_0^L, ..., \_, ..., \_)$, where the underscore "_" symbol denotes a masked element. In this scenario, representation models must predict and fill in the missing state or action components. We directly mask tokens at the input and output levels and do not use masking in attention weights.

**Pre-training objectives.** Next, we introduce the masking patterns investigated in the experimental study. First, the C-GPT masking pattern mimics GPT's masking mechanism and uses causal (backward-looking) attention to predict the next unseen token in RL trajectories. Second, we have the C-BERT masking pattern, derived from BERT's masking mechanism which uses random masks to facilitate diverse learning signals from each trajectory by enabling different combinations. Figure 1 provides a visual representation of the C-BERT and C-GPT masking mechanisms. Third, the MTM masking scheme (Wu et al., 2023) combines random masking (similar to BERT) and causal prediction of the last elements of the trajectory. This latter aims to prevent the model from overly relying on future token information. While MTM operates at the modality level, we adapt it to operate directly on components by masking random tokens within the trajectory and additionally masking a certain proportion of the last tokens. We refer to this method as C-MTM, which stands for component-level MTM. Finally, SMART's training objective encompasses three different masking patterns (Sun et al., 2023): forward-dynamics, inverse-dynamics and masked hindsight control. The training involves adding up the three losses corresponding to the three masking patterns. Similarly, we derive C-SMART, where instead of masking an entire modality at each stage, we mask a random fraction of the tokens within that modality. See Appendix C for additional details.

### 3.4 DOWNSTREAM EVALUATION

In this study, we evaluate the effectiveness of PASTA models in transfer learning from two perspectives. Firstly, we examine the ability of pre-trained models to generate high-quality representations. This evaluation is carried out through probing and parameter-efficient fine-tuning. Secondly, we investigate the capability of pre-trained models to solve new tasks in a zero-shot transfer setting. To accomplish this, we introduce two sets of tasks: **Representation learning** tasks (17) and **Zero-shot transfer** tasks (6), comprising a total of 23 evaluation downstream tasks. These task sets are further

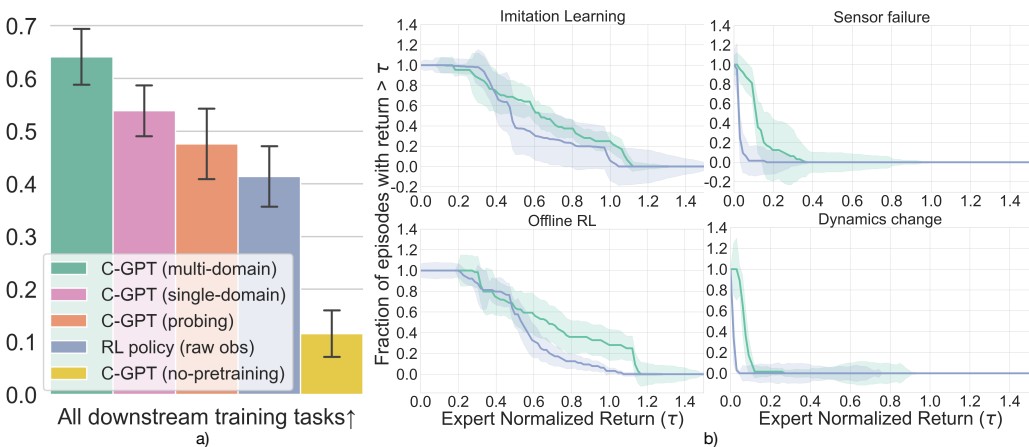

Figure 3: **a)** Evaluation in all downstream training tasks with multi- and single-domain pre-training, no-pretraining, probing, and training from raw observations. Remarkably, C-GPT outperforms all other methods and significantly surpasses policies trained from raw observations, despite having an equivalent number of parameters. ↑ indicates that higher is better. **b)** Performance profile of C-GPT against RL policies trained from raw observations. Shaded areas show the interquartile range over 3 seeds. C-GPT demonstrates higher performance versus policies trained from raw observations.

divided into sub-categories. These categories are designed to provide a general-purpose assessment for pre-trained agents, irrespective of the specific environment or domain. All tasks considered in this study involve either classification or regression.

**Representation learning.**    The representation learning tasks encompass four sub-categories: Imitation Learning, Offline RL, Sensor Failure, and Dynamics Change. We evaluate the quality of raw representations learned by pre-trained agents using probing on these tasks. In this setting, the weights of the pre-trained models are kept fixed, and the embeddings produced by the final attention layer are fed into a single dense layer network. As the expressive power of such networks is limited, achieving good performance is contingent upon the embeddings containing sufficient information. Furthermore, we assess the quality of the produced representations through fine-tuning where the weights of the pre-trained agents are further updated to solve the downstream tasks. For this purpose, we use parameter-efficient fine-tuning, which focuses on updating only a small subset of (or newly introduced) weights representing a small fraction of the total weight volume. Recent studies have shown that these techniques can match or surpass the performance of standard fine-tuning methods while being faster and more memory-efficient. In this study, we use (IA)[3] (Liu et al., 2022b) for fine-tuning, which only updates an average of 0.1% of the total pre-trained weight count.

**Zero-shot transfer.**    The zero-shot tasks are organized into three categories: Action Prediction (AP), Forward dynamics Prediction (FP), and Inverse dynamics Prediction (IP). These categories evaluate the pre-trained models' ability to directly predict states or actions based on trajectory information. Specifically, the prediction problems can be expressed as follows; AP: $(\tau_{t-1}, s_t \rightarrow a_t)$, FP: $(\tau_{t-1}, s_t, a_t \rightarrow s_{t+1})$ and IP: $(\tau_{t-1}, s_t, s_{t+1} \rightarrow a_t)$, where the input to the model is shown on the left side of the parentheses, and the prediction target is shown on the right side. For each category, we examine both component prediction and modality (state or action) prediction.

## 4    EXPERIMENTAL ANALYSIS

In this section, we present the experimental study conducted to examine the impact of pre-training objectives, tokenization, and dataset preparation choices on the generalization capabilities of pre-trained PASTA models.

Table 1: Comparison of models with different tokenization (modality-level, component-level), pre-training datasets (single-domain, multi-domain), and MLP network (RL policy) in the 17 fine-tuned downstream tasks. (max) indicates maximum performance between respectively SMART & MTM and C-SMART & C-MTM. (↑) indicates higher is better and [11] indicates 11 tasks per category.

| Domain | Task | RL policy (raw obs) | Modality-level (max) | Component-level (max) | C-GPT (single-domain) | C-GPT (multi-domain) |
|---|---|---|---|---|---|---|
| HalfCheetah | IL (↑) [1] | 0.719 ± 0.10 | 1.006 ± 0.05 | 1.033 ± 0.02 | 0.974 ± 0.07 | 1.031 ± 0.05 |
| | Off-RL (↑) [1] | 0.903 ± 0.04 | 0.982 ± 0.08 | 1.100 ± 0.03 | 1.031 ± 0.07 | 1.092 ± 0.03 |
| | Sensor failure (↑) [11] | 0.560 ± 0.09 | 0.831 ± 0.08 | 0.871 ± 0.10 | 0.868 ± 0.09 | 0.926 ± 0.07 |
| | Dynamics change (↑) [4] | 0.124 ± 0.05 | 0.175 ± 0.03 | 0.223 ± 0.02 | 0.212 ± 0.02 | 0.248 ± 0.02 |
| Hopper | IL (↑) [1] | 0.571 ± 0.06 | 0.588 ± 0.09 | 1.093 ± 0.10 | 1.132 ± 0.04 | 1.119 ± 0.10 |
| | Off-RL (↑) [1] | 0.412 ± 0.08 | 0.873 ± 0.10 | 1.074 ± 0.01 | 1.169 ± 0.08 | 1.209 ± 0.09 |
| | Sensor failure (↑) [11] | 0.147 ± 0.01 | 0.173 ± 0.04 | 0.355 ± 0.06 | 0.100 ± 0.03 | 0.447 ± 0.03 |
| | Dynamics change (↑) [4] | 0.076 ± 0.00 | 0.139 ± 0.00 | 0.246 ± 0.03 | 0.172 ± 0.03 | 0.258 ± 0.03 |
| Walker2d | IL (↑) [1] | 0.520 ± 0.04 | 0.520 ± 0.05 | 0.519 ± 0.05 | 0.354 ± 0.04 | 0.558 ± 0.03 |
| | Off-RL (↑) [1] | 0.652 ± 0.11 | 0.337 ± 0.05 | 0.376 ± 0.02 | 0.399 ± 0.03 | 0.477 ± 0.05 |
| | Sensor failure (↑) [11] | 0.281 ± 0.03 | 0.206 ± 0.03 | 0.103 ± 0.05 | 0.058 ± 0.02 | 0.322 ± 0.05 |
| | Dynamics change (↑) [4] | 0.000 ± 0.00 | 0.004 ± 0.00 | 0.000 ± 0.00 | 0.000 ± 0.00 | 0.004 ± 0.00 |

## 4.1 EXPERIMENTAL SETUP

**Domains.** To assess the effectiveness of our approach, we select tasks from the Brax library (Freeman et al., 2021a), which provides environments designed to closely match (Freeman et al., 2021b) the original versions found in MuJoCo's environment suite (Todorov et al., 2012). Brax provides significant advantages over MuJoCo, as it offers a highly flexible and scalable framework for simulating robotic systems based on realistic physics. More information about the environments is given in Appendix D.2. The pre-training datasets consist of trajectories collected from three Brax environments: HalfCheetah, Hopper, and Walker2d. Following the protocols used in previous work (Fu et al., 2020; Sun et al., 2023) we trained 10 Soft Actor-Critic (SAC) (Haarnoja et al., 2018) agents initialized with different seeds and collected single- and multi-domain datasets composed of 510 million tokens in total. For details about the pre-training datasets, we refer the reader to Appendix D.3.

Consequently, the 23 downstream tasks presented in Section 3 are set up for each environment resulting in a total of 69 tasks across environments. We introduce multiple environments for pre-training and evaluation (i) to evaluate the reproducibility of our findings across domains and (ii) to study the performance of models pre-trained on the three datasets simultaneously (multi-domains model) compared to single domain models. The implementations of tasks related to sensor failure and dynamics changes need to be adjusted to each domain to account for their specific dynamics. For further details about the implementation of downstream tasks, please refer to Appendix D.4.

**Implementation details.** In this study, we focus on reasonably sized and efficient models, typically consisting of around 10 million parameters. To capture positional information effectively, we incorporate a learned positional embedding layer at the component level. Additionally, we include a rotary position encoding layer following the approach in Su et al. (2021) to account for relative positional information. More implementation details are provided in Appendix B. To convert the collected data (state or action components) into tokens, we adopt a tokenization scheme similar to Reed et al. (2022). Continuous values are mu-law encoded to the range [-1, 1] and discretized into 1024 uniform bins. The sequence ordering follows observation tokens followed by action tokens, with transitions arranged in timestep order.

## 4.2 RESULTS

**Tokenization granularity.** We first examine the impact of tokenization granularity on the generalization performance of the models. We train models using the SMART and MTM training procedures with two granularities: modality-level (predicting at the level of observations and actions) for SMART and MTM and component-level (predicting at the level of observation and action components) for C-SMART and C-MTM. All four models share the same architecture and are trained under identical conditions using the multi-domain dataset. After pre-training, the models undergo fine-tuning for downstream tasks under identical conditions. Figure 2 (a) reports the performance expressed as the mean return normalized by the expert return for each domain. We aggregate the

Table 2: Breakdown of Expert-normalized returns in the sensor failure and dynamics change tasks. (↑) indicates that higher is better.

| Model | Sensor Failure | Dynamics Change |
|---|---|---|
| C-GPT (multi-domain) (↑) | $0.56 \pm 0.04$ | $0.17 \pm 0.02$ |
| C-GPT (single-domain) (↑) | $0.34 \pm 0.05$ | $0.13 \pm 0.02$ |
| RL policy (raw obs) (↑) | $0.33 \pm 0.04$ | $0.07 \pm 0.02$ |
| C-GPT (no-pretraining) (↑) | $0.17 \pm 0.04$ | $0.03 \pm 0.02$ |

results across all domains and task categories: one Imitation Learning task, one Offline RL task, 11 Sensor Failure tasks, and 4 Dynamics Change tasks. Furthermore, Table 1 provides a breakdown of performance for both tokenization techniques across different domains. Overall, we observe that transitioning from modality-level to component-level improves performance for both methods. This improvement is particularly significant for SMART, where the shift nearly doubles the performance for Sensor Failure and Offline RL tasks.

**Masking objectives.** Then, we compare first principles tokenization techniques *i.e.*, masked language modeling (BERT) and next word prediction (GPT) with state-of-the-art transformer RL methods MTM and SMART which incorporate more customized design choices. To adapt BERT and GPT to our problem, we introduce C-BERT and C-GPT. These models are trained using component-level tokenization on the multi-domain dataset under similar conditions as C-SMART and C-MTM, which are the component-level counterparts of MTM and SMART. We systematically fine-tune all models for all downstream tasks and domains. Our findings reveal that C-BERT performs competitively compared to C-SMART and C-MTM across all task categories, as depicted in Figure 2 (a). Additionally, C-GPT exhibits slightly superior average performance for all downstream tasks compared to other masking techniques, as shown in Table 1. This demonstrates that simple and first-principles objectives are sufficient to achieve robust generalization performance.

**Multi-domain representation learning.** Then, we investigate the benefits of pre-training multi-domain representation models using component-level tokens. We synthesize our results using C-GPT as it showed the best performance among the other models in the study. Figure 3 (a) provides a summary of the results aggregated across all domains and downstream tasks, while Figure 3 (b) presents a breakdown of the performance profiles for each downstream task group.

First, we confirm that policies trained using C-GPT outperform policies trained from raw observations with neural networks comprised of an equivalent number of parameters (cf. Appendix D.1). This validates the capability of the model to produce useful representations. We also compare the performance of C-GPT against a randomly initialized model (no-pre-training) with the same architecture and confirm the positive effect of pre-training on the observed performance.

Second, we compare the performance of C-GPT against specialized models trained independently in each environment (C-GPT (single-domain)). To ensure a fair comparison, all models are trained for an equal number of epochs and have the same representation capability (architecture and number of learned parameters). Importantly, the results demonstrate that C-GPT outperforms specialized models in terms of final performance, indicating that C-GPT (multi-domain) learns a more generalizable representation than C-GPT (single-domain). For a detailed breakdown of the results for each downstream task, please refer to panels provided in Appendix A.1.

**Robust representations.** In this section, we focus on resilience to sensor failure and adaptability to dynamics change. These factors play a crucial role in real-world robotics scenarios, where sensor malfunctions and environmental variations can pose risks and impact decision-making processes. We used BC as the training algorithm and during evaluation, we systematically disabled each of the 11 sensors individually by assigning a value of 0 to the corresponding coordinate in the state vector. In Table 2, C-GPT (multi-domain) exhibits higher performance compared to the baselines, demonstrating its enhanced robustness in handling sensor failures. Furthermore, we introduced four gravity changes during the inference phase, and the results reaffirm the resilience of C-GPT (multi-domain) in adapting to dynamics change, thus validating our previous findings.

**Zero-shot predictions.** In this section, we investigate the zero-shot performance of pre-trained models, complementing our previous findings with fine-tuning. We consider an additional set of tasks outlined in Section 3.4, originally introduced in MTM. Notably, Figure 2 (b) shows that C-GPT's errors are comparable to those of specialized models like C-SMART. This suggests a significant alignment between C-GPT's pre-training approach and the inference tasks, even though C-GPT was not explicitly trained to minimize these specific tasks like C-SMART. These results further support the validity of the PASTA methodology, which incorporates first-principles masking patterns, a straightforward objective function, and component-level tokenization. Additionally, we note that C-GPT performs similarly to both C-BERT and C-SMART, except for the inverse dynamics prediction task. This difference is expected since C-GPT is an autoregressive model. Nevertheless, overall performance in Figure 2 (a) suggests that the inclusion of the inverse-dynamics task is not necessary for achieving improved performance across the analyzed downstream tasks.

**Raw representations.** Finally, we examine the representational power of raw pre-trained model embeddings through probing. Probing involves freezing all parameters of the pre-trained neural network during downstream task training. From Figure 3 (a), we observe that, on average, fine-tuning C-GPT outperforms probing by a significant margin. Importantly, the employed parameter-efficient fine-tuning method only introduces a minor average increase of 3.7% in learned parameters, resulting in negligible additional training time.

## 5 DISCUSSION

This paper presents the PASTA study, which focuses on Pretrained Action-State Transformer Agents. The study aims to comprehensively explore self-supervised learning models for RL with downstream training and zero-shot performance evaluation. This study contributes pre-training datasets, a diverse set of 23 downstream tasks, and a comparison of four pre-training objectives and two tokenization techniques. The study was conducted across three continuous control environments to demonstrate the versatility and effectiveness of pre-trained models in various transfer learning scenarios, including probing, parameter-efficient fine-tuning, and zero-shot transfer.

One key finding of this study is the superiority of first-principles objectives over convoluted ones. The results indicate that standard self-supervised objectives, such as random masking or next token prediction, with standard hyperparameters, can match or even outperform more intricate and task-specific objectives designed specifically for RL. This suggests that simpler objectives can serve as a strong foundation for representation learning in RL tasks, simplifying the pre-training process. Additionally, it was observed that tokenizing trajectories based on individual components of the state and action vectors (component-level) rather than directly tokenizing states and actions (modality-level) leads to improved performance in representation models. This observation emphasizes the significance of carefully selecting tokenization strategies to enhance the expressiveness of the learned representations. Furthermore, the study revealed the benefits of pre-training a single model on datasets from multiple domains. Simultaneous pre-training on datasets from different environments resulted in improved performance across all three environments compared to training separate models for each domain. This finding indicates the potential for knowledge transfer and generalization when using diverse pre-training data. For example, the learned representations significantly enhanced the sample efficiency and performance of traditional offline RL algorithms, with an average increase of 23% in final returns compared to the same algorithms using raw inputs. Additionally, the investigation in Section 4.2 highlighted the importance of developing algorithms like C-GPT that can effectively adapt and make decisions in the presence of sensor failures or dynamic changes, ensuring safety and mitigating risks in robotics applications.

Overall, the findings from this study provide valuable guidance to researchers interested in leveraging self-supervised learning to improve RL in complex decision-making tasks. The models presented in this study are lightweight, and the fine-tuning approach involves training fewer than 10,000 parameters, facilitated by PEFT. This feature enables the replication of both pre-training and fine-tuning experiments on readily available hardware, making them accessible to any practitioner. In future work, it is anticipated that further exploration of other self-supervised objectives and tokenization strategies will be conducted. Additionally, expanding the range of downstream tasks will allow for a more comprehensive evaluation of model adaptation and robustness under varying conditions, further enhancing the practical applicability of pre-trained agents in real-world scenarios.

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
