# OpenReview forum: "PASTA: Pretrained Action-State Transformer Agents"
_ICLR.cc/2024/Conference — Submitted to ICLR 2024_

### Official Review · Reviewer_tkJP · 2023-10-29

**Soundness:** 3 good
**Presentation:** 3 good
**Contribution:** 2 fair
**Rating:** 5
**Confidence:** 4

**Summary:**

Several recent papers have explored self-supervised pre-training of transformers on state-action trajectories. These models are then prompted or fine-tuned for imitation learning, RL, forward/inverse dynamics, or state representations. This paper performs a study of different design decisions involved in pre-training transformers for sequential decision making including pre-training objectives (causal prediction, masked prediction, or more complicated strategies used by prior work) and tokenization strategies (modality-level or component-level). The authors conduct experiments for several downstream use cases both with and without fine-tuning. Based on 3 simulated locomotion tasks (HalfCheetah, Hopper, Walker2d), the authors find that component-level tokenization and straightforward causal or masked prediction outperforms prior approaches.

**Strengths:**

The paper makes two original findings: (1) standard causal or random masked prediction objectives outperform more complicated objectives and (2) tokenizing at the state/action component level outperforms tokenizing entire states and actions. In regard to the second finding, component-level tokenization has been used before (e.g Trajectory Transformer [Janner et al 2021]) though I'm not aware of a prior rigorous comparison to modality-level tokenization. Both findings would be of interest to researchers working in this area.

A strength of the paper is the range of downstream use cases that the experiments study, including imitation learning, offline RL, forward and inverse dynamics, zero-shot action prediction, and robustness to sensor failure and dynamics changes.

**Weaknesses:**

My main concerns are in the experimental setup and the presentation of results. The experiments are only performed in 3 domains (Walker, Hopper, HalfCheetah) while prior work often additionally uses other tasks (Ant maze, Adroit, DeepMind Control suite, Franka kitchen). Broad experimental validation is particularly important for a paper that makes general claims about design decisions for an entire class of methods. The presentation of the results is also confusing for the reasons stated below:

- The fine-tuning/probing setup for the representation learning tasks is not well explained. Is some data held out during the dataset generation process and used for fine-tuning in the imitation learning and offline RL settings? For the sensor failure and dynamics change settings, is data generated in the modified MDP and then used for fine-tuning?
- The evaluations of tokenization granularity and pre-training objectives include comparisons to prior work (MTM and SMART). However, the ablation that compares single-domain and multi-domain training is only done with C-GPT (the baseline method). The conclusion of this experiment is that pre-training on multiple domains is better than pre-training on a single domain, however this result has been shown before (e.g. in SMART). A more specific conclusion, like certain masking/tokenization strategies or objectives benefit *more* from multi-domain data, would be of greater interest to the community.
- There should be a better way to organize the results, i.e Figures 2 and 3 and Tables 1 and 2. Currently they contain some duplicated information and it's not clear why the information is grouped in this way. Tables 1 and 2 could be combined since it seems like Table 2 just shows averages of information already in Table 1. Similarly, Figure 3a duplicates some information that's already in Table 1 (C-GPT single and multi domain, "RL policy (raw obs)"), but also adds new methods/ablations (C-GPT probing, no pre-training).
- In Figure 3a there are multiple independent variables: fine-tuning type (probing vs PEFT), and training data (single vs multi domain) but the methods are only labeled with one variable. It's also not clear why all of these methods/ablations are grouped in the same plot.
- In Figure 3a, should "training tasks" be replaced by "evaluation tasks" or just "tasks"? A downstream task is typically not a task that was trained on.
- Figure 3b is never adequately explained. Is each point on the plot the result of pre-training or fine-tuning on datasets with different numbers of successful trajectories?
- Table 1 is confusing because the columns comparing modality-level tokenization and component-level tokenization show the max of SMART and MTM. It would be more clear to keep the masking/pre-training objective the same and only compare the tokenization strategy.
- What is the "RL policy (raw obs) method"? Is this an MLP policy that is pre-trained and fine-tuned in the same way as the other methods (so just a different architecture), or is it trained from scratch on the downstream tasks (hence "raw" observations)?
- Section 3.2 was difficult to understand and would benefit from a rewrite that states the motivation for component-level tokenization more clearly.
- Saying that the evaluation suite contains "23 tasks" seems confusing/misleading. The reader might naturally assume a "task" is one of HalfCheetah, Hopper, or Walker. However, after reading more closely it seems "23 tasks" comes from counting the sensor failure evaluation as 11 tasks (one for each sensor dimension that is removed) and the dynamics change evaluation as 4 tasks (one for each setting of gravity) etc. It seems like the sensor failure and dynamics change evaluations should really be counted as a single "task".
- It would be good to clearly define "modality-level" tokenization/masking somewhere.

**Questions:**

See Weaknesses section.

---

### Official Review · Reviewer_SZP5 · 2023-10-29

**Soundness:** 2 fair
**Presentation:** 2 fair
**Contribution:** 2 fair
**Rating:** 3
**Confidence:** 4

**Summary:**

This paper present a study on different design choices for Transformer-based policy architectures for various downstream applications including behavior cloning and offline RL. The goal is to investigate what components of the pretraining pipeline is most important for effective representation learning for fast downstream adaptation. Their study highlights a few interesting findings that could be useful for researchers looking to pretrain Transformer models for their RL tasks. For example, objectives like random masking or next token prediction are better than hand-crafted objectives. Additionally, tokenization granularity is important to consider during pretraining.

**Strengths:**

Study is comprehensive and considers a variety of factors in pretraining including tokenization, pretraining objective, datasets, and downstream applications.

- Interesting idea to break down the state representation into modular components (such as robot's morphology) depending on the environment which seems to improve performance over vanilla input representations.

- Large set of downstream tasks evaluating both the learned representations and zero-shot transfer (23 total tasks). Tasks including probing, parameter-efficient FT, and zero-shot predictive tasks.

**Weaknesses:**

Most of the results that the paper finds are seemingly intuitive (e.g. more diverse data is better for generalization, next-token prediction is a good objective...)

The main improvement in performance in Figure 2 seems to come from the component level token representation although that is not highlighted as the main contribution of this work. Additionally, modular token inputs have been explored in other

C-MTM objective seems to be comparable to GPT / BERT objectives in Figure 2, contrary to the claim that simple objectives are better than "task-specific" objectives. Actually, the Mean Absolute Error is better for C-MTM compared to C-BERT and C-GPT.

Results from Figure 3 only show that training with more diverse data is better and that a Transformer-based policy is better than MLP policy. This result is not new.

In Table 1, there is no or negative performance change in the Walker2D task when using component level inputs. There is no explanation for this in the paper.

**Questions:**

Table 1. What do the values mean? Are these normalized returns?

In Table 2, do you keep the dataset size for training constant or does the multi-domain dataset strictly have more data samples than the single-domain. If so, can the performance improvement be attributed to training on a larger dataset?

Decision Transformers and similar works that use autoregressive Transformers for learning from offline data seem relevant. Have you considered including results for those models in the study?

What is the purpose of evaluating both behavior cloning and offline RL if the core investigation is about representations? I don't think having results for both method provides any new insight.

**Details Of Ethics Concerns:**

There is no mention in the paper of possible ethical concerns introduced by the results of the study. There is a slight mention that methods like C-GPT can lead to robust representations that can help to ensure safety in robotic applications.

---

### Official Review · Reviewer_tuMZ · 2023-10-30

**Soundness:** 2 fair
**Presentation:** 3 good
**Contribution:** 2 fair
**Rating:** 3
**Confidence:** 3

**Summary:**

This work conducts a study of action-state transformer pre-training for reinforcement learning. In particular, it compares design choices such as tokenization of the space, pre-training objectives and training settings. Training settings include imitation learning, offline RL as well as changing environment conditions. The work also investigates a componentwise tokenization technique. The results indicate that simple masking objectives can be sufficient and component-wise tokenization might be useful in certain settings.

**Strengths:**

Motivation
* The work is well motivated. It is important to have a broad understanding of a certain class of approaches and consolidate knowledge across various approaches and domains.

Challenging Existing Results
* I think there is general value in replicating existing results under separate implementations and checking for robustness to implementation changes. There is scientific value in challenging existing results and summarizing overall insights across methods in comparison studies.

Structural clarity
* The paper has an easy to follow structure and the line of thought throughout reading it was clear. Even though the paper could be a little more concise in some places (abstract, conclusion) I think it does a good job of communicating the overall story.
* Figures and tables are mostly explained decently. Minor exceptions are highlighted in the textual clarity suggestions.

Contextualization with prior work
* The work is well contextualized with respect to prior work. It is clear why the baselines in the work were chosen and relevant literature has been highlighted.

**Weaknesses:**

Algorithmic novelty
* The main algorithmic contribution seems to be component-wise tokenization but there is no information on how that is done or ablations on it. It is not clear what tokenization works best in which environments and if there are particular patterns to how one should tokenize. It is also not clear if this tokenization might differ across simulator domains. The work at some points motivates the components as a type of logical decomposition (bi-pedal example) but this strain is not followed up on or investigated. To study something like this, environments with compositional state-spaces such as those in the CompoSuite benchmark might be of interest.
* Other insights such as masking patterns have already been studied in cited publications which questions the novelty of that part of the study. However, results seem to contradict previous work.

Experimental evaluation
* Some of the design choices like network structures, positional encoding are not explained well (see Q2, Q3, Q4).
* The claim that component-wise tokenization works well could be better supported by additional BERT and GPT-style masking ablations. It is not clear to me why these were omitted since they seem to be simple baselines to strengthen the claim.
* The claim that multi-domain pre-training is helpful seems not sufficiently supported (see Q6).
* While there is a variety of settings, the paper lacks variety in environments. All environments are environments that arguably have similar dynamics systems and state-action representations. An extensive study of the subject might want to consider other general domains, e.g. video games with visual representations (Atari, Procgen), robotic arm simulators (Meta-world style environments) or text-based environments (nethack).

Comparability to prior work
* The work builds its own datasets instead of using datasets from the literature that would enhance comparisons. Also, when re-implementing baselines there is a certain danger that important details that make certain methods work are omitted. It is not clear to me why the results presented in this study differ from those in previous work (see Q5, Q6).

Textual clarity suggestions
* The abstract is relatively long and does not contain any information about the findings of the paper. It might make sense to shorten the intro of the abstract and add a summary of the experimental results.
* Missing citations or evidence for claim on P4, “breaking down leg movements into thigh and shin components can facilitate the learning of more stable and efficient walking or running”. It seems like this would be something to show.
* It is not exactly clear what PASTA is referring to. Sometimes it’s the PASTA study and sometimes the PASTA methodology.
* Table 1 is very hard to read and might benefit from some highlighting. It is not quite clear what the max operator indicates and it took me a while to figure out what columns I am supposed to be comparing since they seem to be testing different things and are not fully defined in the caption.
* Table 2 up-arrows should probably be on “Sensor Failure” and “Dynamics Change” rather than the methods.
* I am having trouble making out which tasks are seen and which are unseen to the agent. It might make sense to highlight this better.
* The conclusion just repeats a lot of text that was already mentioned. It could probably be more concise to focus on the insights that were obtained from the study to gain some space.

My overall conclusion is that this paper is decently written but tries to do two things at the same time and does not make a strong enough case for either. First, the introduction of component-wise tokenization is the algorithmic novelty that is not really studied or explained properly. Second, for a comparison study, the lack of variety in simulators leads to some claims being weaker than they could be. I think this work has the promise to be quite valuable but for now, I am recommending rejection. I encourage the authors to focus on one direction and put a larger emphasis on the study of either the compositional encoding or the attempt to provide a general overview of existing approaches across domains/simulators.

**Questions:**

1.) Can you elaborate why simpler objectives are referred to as first principles? Do they come with some guarantees derived from first principles?

2.) In the fine-tuning settings, do the policy heads contain any intermediate layers? If yes, where is the difference between introducing a limited number of parameters to the pre-trained model and only pre-training policy heads?

3.) It is not clear what training on “raw obs” means for the RL policies and whether these policies are pre-trained or not. If they are not pre-trained, for how long are they trained? Do they have access to the same amount of data as the transformer models?

4.) Why is there a positional encoding on the component level? It seems that the order of states within one component should not matter.

5.) The results of this work seem to contradict the findings in other published studies. For instance, turning off certain objectives in MTM/SMART was shown to have a negative impact on performance in the original publications. Why do you think that is? Where are the differences that can be highlighted to explain the discrepancies? Also, why did you choose not to use original implementations for baselines?

6.) Can you elaborate why you chose to not use existing offline RL datasets? Choosing datasets from the literature might increase comparability. Also, what is the performance of the final agents in the last 20% of trajectories that are collected?

7.) How can we be sure that the increase in performance from multi-domain to single-domain does not come from pre-training on more data rather than the variety of the environments?

---

### Official Review · Reviewer_vct6 · 2023-11-01

**Soundness:** 3 good
**Presentation:** 3 good
**Contribution:** 3 good
**Rating:** 6
**Confidence:** 4

**Summary:**

The authors empirically study the pretraining of transformer-based policy learning. They design extensive experiments across various axes of pretraining, e.g., tokenization, pretraining objective, single/multiple domain samples and finetuning methods. From the large-scale experiments, authors provide some insights for policy pretraining.

**Strengths:**

1.	The authors conduct extensive experiments on 69 tasks in total and the results are solid.

2.	There are some interesting conclusions, especially about the tokenization method, which is insightful for policy pretraining.

**Weaknesses:**

My concerns are mainly about the experiment setup and comparison fairness. Please refer to the Questions part for the details.

**Questions:**

1. For the data collection, in Appendix D3, I find authors select the 20% latest trajectories of size 1000. So the pretraining dataset is some “expert” trajectories. Does it mean that there are some assumptions or requirements on the data quality but you didn’t mention this in your paper? Will this bring some bias into your empirical study? Please also discuss whether and how your method can be generalized to play data.

2. In Section 3.2, would you please clarify how you design the positional embeddings on time and component axes? Do you use different modality embeddings for state and action respectively?

3. I understand the authors want to cover more experimental setups and gave up on more diverse environments. But only taking Hopper, HalfCheetah and Walker2d as testbeds is not enough. The experimental results might be biased to these locomotion tasks. I expect there will be some results on manipulation or game tasks.

4. The main story in Introduction is about pretraining for reinforcement learning. However, for downstream tasks, authors only select imitation learning and offline RL. I’m willing to see some results on standard online RL.

5. In Section 4.2 “Multi-domain representation learning”, authors compare the results of pretraining the model on multi-domain and single-domain tasks. But the numbers of pretraning samples are different. I think this comparison is unfair.

---

### Author Response · Authors · 2023-11-21
**Author Response**

We would like to thank all the reviewers for their valuable insights and feedback. We will carefully consider all comments and address each of them to the best of our ability.

---

### Meta-Review · Area_Chair_B14Y · 2023-12-09

**Metareview:**

Most reviewers recommended rejection, on the basis of the soundness of the experimental design and algorithmic novelty. Although they gave detailed reviews, the authors didn't provide any rebuttals or revisions, which is a shame because the reviews all had positive things to say about it and commend its presentation. I hope these comments will be helpful for future submissions.

**Justification For Why Not Higher Score:**

Reviewers raised concerns about the experimental design, evaluation, and lack of focus in its presentation, which indicate this work doesn't yet meet the bar for acceptance. As there was no rebuttal, I think it's appropriate to reject.

**Justification For Why Not Lower Score:**

NA

---

### Decision · Program_Chairs · 2024-01-16

Reject